# NeRF-IBVS: Visual Servo Based on NeRF for Visual Localization and Navigation

**Yuanze Wang**[1,2*]   **Yichao Yan**[1*]   **Dianxi Shi**[1,2†]   **Wenhan Zhu**[1]    **Jianqiang Xia**[2,3]
**Tan Jeff**[3]   **Songchang Jin**[2]   **Ke Gao**[4]   **Xiaobo Li**[4]   **Xiaokang Yang**[1]

[1]MoE Key Lab of Artificial Intelligence, AI Institute, Shanghai Jiao Tong University
[2]Intelligent Game and Decision Lab (IGDL), Beijing, China
[3]Tianjin Artificial Intelligence Innovation Center [4]Alibaba Group
{yz.wang,yanyichao,zhuwenhan823,xkyang}@sjtu.edu.cn
dxshi@nudt.edu.cn,jianqiang.xia@foxmail.com,jsc04@tsinghua.org.cn
{xiaobo.lixb,gaoke.gao}@alibaba-inc.com

## Abstract

Visual localization is a fundamental task in computer vision and robotics. Training existing visual localization methods requires a large number of posed images to generalize to novel views, while state-of-the-art methods generally require ground truth 3D labels for supervision. However, acquiring a large number of posed images and 3D labels in the real world is challenging and costly. In this paper, we present a novel visual localization method that achieves accurate localization while using only a few posed images compared to other localization methods. To achieve this, we first use a few posed images with coarse pseudo-3D labels provided by NeRF to train a coordinate regression network. Then a coarse pose is estimated from the regression network with PNP. Finally, we use the image-based visual servo (IBVS) with the scene prior provided by NeRF for pose optimization. Furthermore, our method can provide effective navigation prior, which enables navigation based on IBVS without using custom markers and the depth sensor. Extensive experiments on 7-Scenes and 12-Scenes datasets demonstrate that our method outperforms state-of-the-art methods under the same setting, with only 5% to 25% training data. Furthermore, our framework can be naturally extended to the visual navigation task based on IBVS, and its effectiveness is verified in simulation experiments.

## 1 Introduction

The estimation of camera positions and orientation based on a query image is a fundamental task in computer vision. It has wide applications in various fields such as robotics, virtual reality, and autonomous driving. Existing works that tackle this task can be generally divided into two categories. The first line of approaches are structure-based [6, 28, 31, 37, 46], which first establish correspondences between 3D points and 2D pixels, and then compute camera pose with PnP [16] solver utilizing RANSAC [12]. These methods typically achieve satisfactory results at the cost of a complex pipeline. The other line of approaches is regression-based [2, 3, 40, 21, 42, 41], which either regress the 3D coordinates or directly regress the absolute pose. These methods yield a more compact framework and have fast inference performance. All the above methods require a substantial number of posed images (generally **thousands** of posed images per scene) to guarantee the generalization ability to novel viewpoints. Moreover, state-of-the-art methods [6, 4, 21] also require dense ground truth 3D labels for supervision, such as depth and 3D model. However, acquiring a large number

---

*These authors contributed equally to this work.
†Corresponding authors.

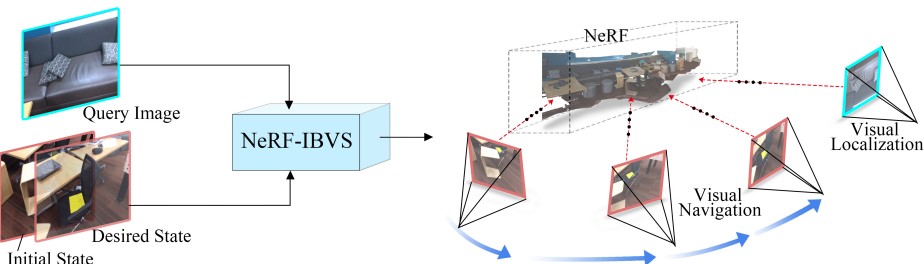

Figure 1: Demonstration of the proposed NeRF-IBVS. Our method can achieve visual localization based on a single query image. Moreover, our method can obtain navigation prior after processing the images of the initial state and desired state, which can better enable IBVS-based navigation.

of posed images and dense 3D labels is challenging and costly. It naturally raises a question: *is it possible to design a framework that achieves accurate localization results while only requiring a small number of posed images ?*

Recent advances in Neural Radiance Fields (NeRF) [25] provide a promising solution. Trained from **hundreds** of posed images, NeRF can render realistic images and give coarse depth maps for novel viewpoints as prior knowledge of 3D scenes. Intuitively, this prior knowledge can assist visual localization methods in generalizing to novel viewpoints. Current NeRF-based visual localization methods [10, 11, 26] only employ NeRF for data augmentation purposes, which still requires a substantial number of images for training. The estimated pose will be coarse when using a few data. Considering this issue, it would be desirable if the coarse pose in NeRF could be further refined to meet the need for visual localization.

In this paper, we propose a novel visual localization framework called NeRF-IBVS, which achieves accurate localization while using only a few posed images. NeRF-IBVS adopts a coarse-to-fine paradigm to estimate the pose of the query image. In the coarse stage, we first train a NeRF with hundreds of posed images, which is significantly fewer than typical visual localization methods. Then, we use the same posed images with coarse pseudo-3D labels provided by NeRF to train a coordinate regression network. The coarse pose is estimated from the regression network with PNP. In the refining stage, we design an optimization algorithm to effectively optimize the coarse pose. Motivated by the image-based visual servo (IBVS) [9], which navigates the camera from the initial pose to the desired pose using coordinate error and depth of correspondences between the current image and the target image. Therefore, we first use NeRF to render the image and depth corresponding to the current coarse poses. Then, we establish the correspondences between the rendered image and the query image and query the approximate depth of the correspondences based on the rendering depth map. Finally, the correspondences with depth are used to launch IBVS to guide pose optimization.

Due to rendering quality errors, directly applying IBVS to NeRF-rendered images and depth is not feasible. Moreover, IBVS requires non-collinear correspondences during 6-degree-of-freedom navigation. In order to achieve robust pose optimization, we design a correspondence selection algorithm using spatial information provided by NeRF and the coordinate regression network. To speed up the pose optimization, we further design a strategy that reduces the rendering frequency of NeRF. Further, the effective 2D and 3D correspondences obtained by NeRF-IBVS can be used as the navigation prior to enhance IBVS-based navigation. Specifically, this navigation prior can enable IBVS-based navigation without using custom markers and the depth sensor. This advantage greatly expands the application scope.

To the best of our knowledge, NeRF-IBVS is the first method that can accomplish both visual localization and visual navigation based on IBVS as shown in Figure 1, providing a novel idea for pose optimization by controlling camera motion. Employing only 5% to 25% of the posed images required by other localization methods on 7-Scenes dataset and 12-Scenes dataset, our method outperforms state-of-the-art methods without using ground truth 3D labels and has comparable performance with methods using ground truth 3D labels. Furthermore, our method can enable visual navigation based on IBVS without using custom markers and the depth sensor, which greatly expands the application scope of IBVS-based navigation. We also verified the effectiveness of visual navigation in a simulation environment.

## 2 Related Work

**Visual localization.** Structure-based methods [6, 28, 31, 37, 46] use complex pipelines to achieve precise localization. Firstly, correspondence is established between 3D coordinates and 2D pixels. Secondly, a camera pose is computed through a PnP solver with RANSAC. The first stage requires a complex pipeline consisting of image retrieval, feature extraction, and matching. To achieve a more compact frame compared to structure-based methods. In recent years, regression-based localization methods [2, 3, 40, 21, 42, 41, 24] attract increasing attention, which either regresses the 3D coordinates or directly regress the absolute pose. Specifically, coordinate regression methods, such as DSAC++ [7] and HACNet [21] use neural networks to learn the first stage of the structure-based methods, which directly regress 3D scene coordinates from an image. These coordinates regression methods typically use ground truth 3D labels as supervision to achieve better localization performance. Unlike coordinate regression, absolute pose regression methods [19, 40, 2] generally use only posed images to train the neural network. MS-Transformer [32] and AtLoc [41] use attention mechanisms to guide the regression process and have substantially improved the performance of absolute pose regression methods. Recently, some methods [10, 11, 26] use NeRF as data enhancement to improve the performance of visual localization methods. The above methods require a substantial number of posed images to guarantee the generalization ability to novel viewpoints. Our method can achieve accurate localization while using only a few posed images compared to other localization methods.

**Robotics with neural radiance fields.** Recently, the neural radiance field (NeRF) [25] has shown great potential in robotics applications [22, 17, 15]. iNerf [44] fixes the network weights of NeRF and directly optimizes the camera pose parameters by computing the backpropagation of the photometric loss. However, this method requires an initial pose near the ground truth pose. Methods based on photometric loss tend to fail when blur and artifacts appear in the rendered images. iMAP [36] and NICE-SLAM [47] are a pose estimator in the SLAM setting that requires continuous sequential data. BARF [23] optimize camera poses during the training of NeRF. Therefore they cannot estimate the pose of a single query image. Moreover, NeRF can provide coarse scene geometry, which is useful for navigation tasks. NeRF-nav [1] designs a smooth and collision-proof navigation strategy based on the density provided by NeRF. RNR-Map [20] achieves precise visual navigation performance based on NeRF, but is limited to three degrees of freedom and requires RGBD pictures and odometer information. Our method can accomplish both visual localization and 6 degrees of freedom navigation based on visual servo and using only RGB images.

## 3 Preliminaries

**NeRF.** NeRF uses MLP network $F_\theta$ to encode the 3D scene, which predicts the RGB color $\mathbf{c} \in \mathbb{R}^3$ and volume density $\sigma \in \mathbb{R}$ for each input 3D point $\mathbf{p} \in \mathbb{R}^3$ and its view direction $\mathbf{d} \in \mathbb{R}^3$. It can be summarized as $F_\theta(\mathbf{p}, \mathbf{d}) = (\mathbf{c}, \sigma)$, where $\theta$ is the parameters of MLP network. Then, NeRF uses volume rendering techniques to generate realistic rendered images $C(r)$ and coarse depth maps $D(r)$ from any viewpoint:

$$C(\mathbf{r}) = \sum_{i=1}^{M} T_i(1 - exp(-\sigma_i\delta_i))c_i, \ D(r) = \sum_{i=1}^{M} T_i(1 - exp(-\sigma_i\delta_i))t_i, \tag{1}$$

where $\mathbf{r}$ and $M$ indicates the rendering ray and the number of points along the ray, $c_i$ and $t_i$ indicates the color and distance of the samples in the rendering ray, $T_i = (-\sum_{j=1}^{i-1} \sigma_j\delta_j)$ indicates the accumulated transmittance from the near bound of the rendering ray to $t_i$, and $\delta_i = t_{i+1} - t_i$ is the distance between adjacent samples in rendering ray.

**IBVS.** The aim of Image-Based Visual Servoing (IBVS) is to control the camera to move toward the desired pose based on vision information while minimizing the coordinate error of image features between the current image and the target image. Correspondences are used as image features in this paper. To achieve this goal, IBVS first calculates the desired image coordinate velocity of correspondences based on the coordinate error. Then, the Jacobi matrix is constructed based on the correspondences, which establishes the relationship between the image coordinate velocity and the camera velocity. Finally, the desired camera velocity to control the camera motion towards the target is solved based on the Jacobi matrix and the desired image coordinate velocity. The specific details of the IBVS are presented in the supplemental materials.

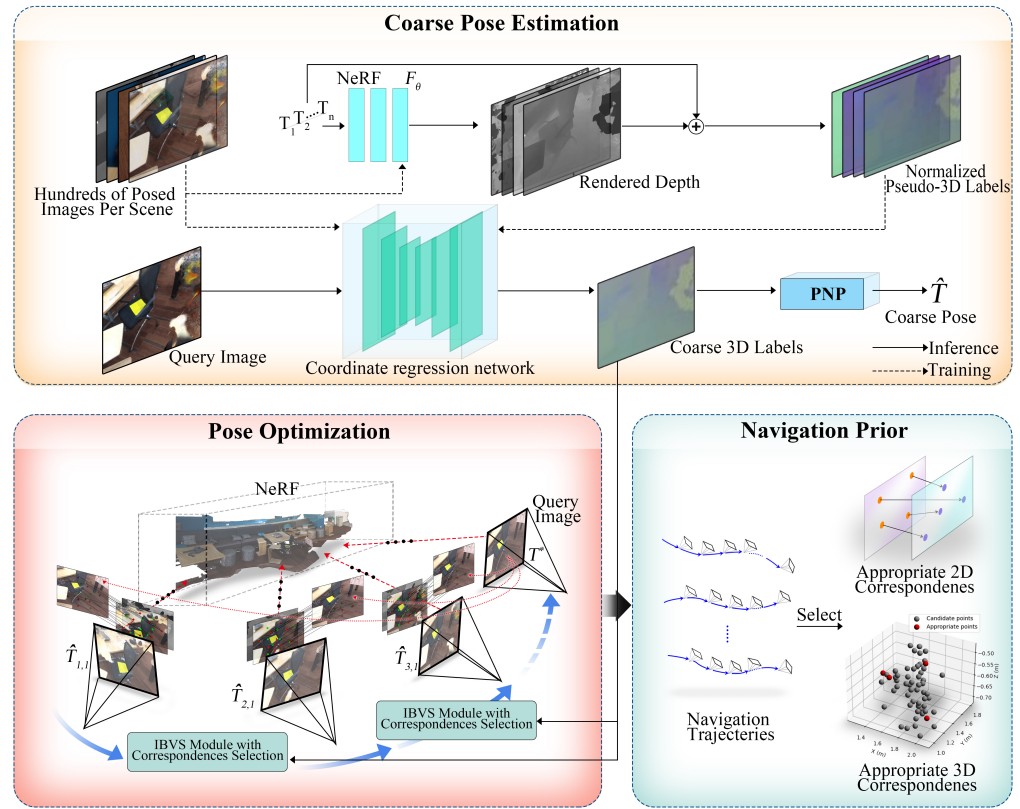

Figure 2: The pipeline of our method, NeRF-IBVS. In the coarse pose estimation, we first train a NeRF with hundreds of posed images, which is significantly fewer than typical visual localization methods. Then, we use the same posed images with coarse pseudo-3D labels provided by NeRF to train a coordinate regression network. The coarse pose is estimated from the regression network with PNP. In the pose optimization, the correspondences with depth between the rendered image and the target image are obtained to launch IBVS to guide pose optimization. Moreover, our method can provide effective navigation prior, which enhances navigation based on IBVS.

## 4 Method

Given a few posed images compared to the typical localization method, we aim to achieve accurate localization. To achieve this, we first use a few posed images with pseudo-3D labels provided by NeRF to train a coordinate regression network. Then the coarse pose is estimated from the regression network with PNP. Finally, we introduce IBVS to gradually refine the coarse pose by leveraging the 3D scene prior provided by NeRF. Furthermore, our method can provide effective navigation prior, which enhances navigation based on IBVS. Figure 2 demonstrates the overall pipeline.

### 4.1 Coarse Pose Estimation

We use a few posed images with pseudo-3D labels provided by pre-training NeRF to train a coordinate regression network. Then the coarse pose is estimated from the regression network with PNP. Specifically, we convert the pixel coordinates $\mathbf{p}$ of training images into pseudo-3D labels $\mathbf{P_w}$ using depth $D$ provided by NeRF:

$$\mathbf{P_w} = D\mathbf{T}^{-1}\mathbf{K}^{-1}\mathbf{p}, \tag{2}$$

where $\mathbf{K}$, $\mathbf{T}$ denote the internal parameter and pose of camera. Following common settings [43], these pseudo-3D labels $\mathbf{P_w}$ are then normalized to fit within the range of a unit cube. Then, we use the advanced coordinate regression component of GDR-Net [42] as our coordinate regression network, which is trained by pseudo-3D labels $\mathbf{P_w}$. To estimate coarse pose, the coordinate regression network

processes the query image to generate 3D coordinates and establishes a 2D-3D correspondence. This 2D-3D correspondence is then fed into PnP [16] to determine the coarse pose $\hat{\mathbf{T}}$ of the query image. We empirically found that the lack of multi-view constraints in the boundary regions of the scene leads to a significant depth error in image edges. To mitigate the impact of the rendering quality error, we exclude the edge areas in the image of our pseudo-3D labels $\mathbf{P_w}$ when training our coordinate regression network.

## 4.2 Pose Optimization

To optimize the coarse poses, our method views pose optimization as the process of navigating the camera from the coarse pose towards the target pose using scenes prior provided by NeRF.

### 4.2.1 IBVS Module

**IBVS initialization using NeRF.** To start pose optimization, we first initialize IBVS using NeRF. This process mainly initializes the Jacobian matrix $\mathbf{L}$ of IBVS using the image coordinates and depth of correspondences, which establishes the relationship between the velocity $\mathbf{v}_c$ of the camera and the velocity $\dot{\mathbf{s}}$ of image coordinates:

$$\dot{\mathbf{s}} = \mathbf{L}\mathbf{v}_c, \tag{3}$$

where the $\mathbf{L}$ corresponds to per correspondence can be formulated as:

$$\mathbf{L} = \begin{bmatrix} -\frac{1}{Z_c} & 0 & \frac{x}{Z_c} & xy & -(1+x^2) & y \\ 0 & -\frac{1}{Z_c} & \frac{y}{Z_c} & 1+y^2 & -xy & -x \end{bmatrix}, \tag{4}$$

where $x, y$ and $Z_c$ denote image coordinates and depth of the correspondence in the currently rendered image. To initialize the Jacobian matrix $\mathbf{L}$, we first utilize NeRF to render an image $C_r$ and depth map $D_r$ that corresponds to the current coarse pose $\hat{T}$. Then the off-the-shelf correspondences establishment algorithm, such as SuperPoint [14] with SuperGlue [29], is employed to extract the pixel coordinate $(\mathbf{m_q}, \mathbf{m_r})$ of correspondences between the query image $C_q$ and the rendered image $C_r$. The depth $Z_c$ of correspondences $\mathbf{m_r}$ can be obtained by querying the rendered depth $D_r$. Image coordinates $\mathbf{n_r} = (x, y)$ in the rendered image can be obtained using the pixel coordinates $\mathbf{m_r}$ and the internal parameter $\mathbf{K}$ follow [9]. Finally, $\mathbf{n_r}$ with depth $Z_c$ are used to initialize $\mathbf{L}$.

**Optimization iteration.** After the IBVS initialization, we started IBVS iteration to navigate the camera from the coarse pose toward the target pose. At each iteration, IBVS needs to calculate the velocity $\mathbf{v}_c$ of the camera to update the pose. To control the 6 DOF velocity $\mathbf{v}_c$ of the camera, four non-collinear correspondences are selected and the velocity $\dot{\mathbf{s}}$ of the image coordinates are set to $-\lambda(\mathbf{n_q} - \mathbf{n_r})$ follow [9]. According to Equation (3), we can obtain the camera velocity $\mathbf{v}_c$ as:

$$\mathbf{v}_c = -\lambda\mathbf{L}^+(\mathbf{n_q} - \mathbf{n_r}), \tag{5}$$

where $\lambda, \mathbf{L}^+$, and $\mathbf{n_q}$ denote proportional factor, generalized inverse matrix, and image coordinates in the query image. Then, we update the current pose by integrating the camera velocity $\mathbf{v}_c$ during unit iteration to obtain the variation of pose $\Delta\mathbf{T}(\mathbf{v}_c(t))$. The coarse poses are updated as follows:

$$\hat{\mathbf{T}}(t + 1) = \hat{\mathbf{T}}(t)\Delta\mathbf{T}(\mathbf{v}_c(t)). \tag{6}$$

The IBVS iteration proceeds until either the pixel coordinate error of correspondences is less than a certain threshold or the maximum number of iterations is reached.

Theoretically, each iteration of IBVS requires NeRF rendering to update the parameters of the Jacobian matrix $\mathbf{L}$. However, utilizing NeRF rendering for each IBVS iteration is highly time-consuming and impractical. Therefore, we design a new fast iteration strategy that reduces the rendering frequency of NeRF. We first calculate the 3D coordinates of correspondences using the depth information provided by NeRF, while assuming the 3D coordinates of the correspondences are precise. At each iteration, We project the 3D coordinates of correspondences to the image plane by using the updated poses. This projection can be an approximate substitute for the correspondence in the rendered image. In this way, we skip the time-consuming NeRF rendering process, and NeRF rendering is only executed at the initialization of IBVS.

Due to rendering errors of NeRF, the assumption that the 3D coordinates of correspondences are accurate is violated, so accumulation errors will occur. Accumulated errors may lead to the optimization terminating with a large error, In order to eliminate accumulated errors, we render the image

again based on the most recently updated poses to relaunch IBVS iterations. Typically, the IBVS iterations are launched no more than four times to obtain a precise localization.

The first IBVS iterations launch aims to move the camera a significant distance toward the target pose, while the subsequent IBVS iterations launch only fine-tune the pose around the target pose. To improve the optimization efficiency, We dynamically set the number of iteration steps. Specifically, the number $N(i)$ of iterations steps in the $i$th IBVS iterations launch is set dynamically based on coordinate error of correspondences:

$$N(w(i)) = \begin{cases} N_1, & w(i) > 1 \\ \frac{N_1}{3}, & w(i) < \frac{1}{3} \\ w(i)N_1, & \text{otherwise} \end{cases}, \quad \text{where } w(i) = \frac{\|\mathbf{n_r}^i - \mathbf{n_q}^i\|_2}{\|\mathbf{n_r}^1 - \mathbf{n_q}^1\|_2}, \tag{7}$$

where $N_1$ and $\mathbf{n_r}^i, \mathbf{n_q}^i$ denote the fixed number of iteration steps in the first IBVS iteration launch and image coordinates of correspondences in the $i$th IBVS iterations launch, respectively.

### 4.2.2 Correspondence Selection

The Jacobian matrix $\mathbf{L}$ of IBVS is highly sensitive to the selection of the correspondences, and it requires accurate and non-collinear correspondences to update parameters during iteration. Due to rendering errors of NeRF, which may cause inaccurate correspondences. In order to increase the robustness of IBVS, we designed a correspondence selection algorithm to select appropriate correspondences for IBVS. We first query coarse 3D coordinates estimated by the coordinate regression network to obtain 3D coordinate $\hat{\mathbf{P}}_\mathbf{q}$ of correspondences $\mathbf{m_q}$ in the query image. Then, we use coarse pose $\hat{\mathbf{T}}$ to project the 3D coordinate $\hat{\mathbf{P}}_\mathbf{q}$ onto the rendered image to obtain image coordinates $\hat{\mathbf{n}}_\mathbf{r}$. Finally, we can obtain the coordinate distances of correspondences $d = \|\hat{\mathbf{n}}_\mathbf{r} - \mathbf{n_r}\|_2$. Since the coarse pose $\hat{\mathbf{T}}$ and 3D coordinates $\hat{\mathbf{P}}_\mathbf{q}$ near the target, the accurate correspondences generally have a smaller distance. Therefore, we empirically set a threshold $\tau$ for the coordinate distance $d$ to filter out a large number of outliers. Finally, we use RANSAC [12] to further select accurate and non-collinear correspondences. Please see the supplementary material for more details.

### 4.3 Visual Navigation

NeRF-IBVS can provide navigation prior which enhances IBVS-based navigation. To enable IBVS navigation, it is necessary to use correspondences that are non-collinear and remain in the camera's field of view during navigation. However, acquiring such correspondences is challenging, current methods [18, 45, 34] usually rely on custom markers which limit the applications of IBVS. The navigation prior provided by our method can solve this problem. Specifically, We first use NeRF-IBVS to locate the initial pose of the navigation task. Then the initial pose is viewed as a coarse pose to start the NeRF-IBVS locating the target state of the navigation task. After that, the correspondence selection algorithm can obtain accurate and non-collinear correspondences and the IBVS module can obtain navigation trajectories in the image plane. The current correspondences are not yet guaranteed to remain in the camera's field of view during navigation. Benefiting from the navigation trajectory, the appropriate 2D correspondences that fall outside of the image plane are automatically filtered out. It ensures that the correspondences selected remain in the camera's field of view during navigation. We backproject 2D correspondences based on depth provided by NeRF to obtain coarse 3D coordinates as 3D correspondences. During the navigation iteration, the 3D correspondences are projected using the current pose to obtain the depth prior and the image coordinates prior. The image coordinates prior can assist in locating and associating the 2D correspondences in the current image, which enables IBVS-based navigation without custom markers. Moreover, the depth prior can be used to update the parameters of the Jacobian matrix in each iteration, which enables IBVS-based navigation without using a depth sensor. Although there is a small error in the depth obtained by projection, IBVS has the advantage of being robust to small depth errors[13].

## 5 Experiments

In this section, we first compare the proposed NeRF-IBVS with state-of-the-art visual localization methods. Furthermore, our method extended to IBVS-based navigation without using custom markers and the depth sensor, and its effectiveness is verified in simulation experiments.

## 5.1 Visual Localization

**Datasets.** We conduct experiments on two public indoor benchmark datasets. The 7-Scenes [35] contains seven indoor scenes recorded by a KinectV1 camera, the data includes RGB-D images, poses, and ground truth 3D models. The 12-Scenes [39] contains twelve indoor scenes with RGB-D images and poses. The recorded environments of the 12-Scenes are significantly larger than those in the 7-Scenes. We use fewer posed images compared to baseline localization methods, and the specific number of training data for each scene is configured as shown in Table 1 and Table 2.

Table 1: The number of training data on the 7-Scenes dataset.

|  | Chess | Fire | Heads | Office | Pumpkin | Kitchen | Stairs | All |
|---|---|---|---|---|---|---|---|---|
| Baseline | 4000(100%) | 2000(100%) | 1000(100%) | 6000(100%) | 4000(100%) | 7000(100%) | 2000(100%) | 26000(100%) |
| Our | **260(6%)** | **280(14%)** | **240(24%)** | **300(5%)** | **320(8%)** | **350(5%)** | **340(17%)** | **2090(8%)** |

Table 2: The number of training data on the 12-Scenes dataset.

|  | Kitchen1 | Living1 | Bed | Kitchen2 | Living2 | Luke | Gates362 | Gates381 | Lounge | Manolis | Office5a | Office5b | All |
|---|---|---|---|---|---|---|---|---|---|---|---|---|---|
| Baseline | 744(100%) | 1036(100%) | 868(100%) | 768(100%) | 725(100%) | 1370(100%) | 3536(100%) | 2950(100%) | 925(100%) | 1613(100%) | 1001(100%) | 1391(100%) | 16927(100%) |
| Our | **185(25%)** | **170(16%)** | **215(25%)** | **190(25%)** | **180(25%)** | **340(25%)** | **350(10%)** | **295(10%)** | **230(25%)** | **270(17%)** | **250(25%)** | **280(20%)** | **2955(17%)** |

**Baseline.** Baseline methods can be classified into two main categories based on whether or not using ground truth 3D labels. Specifically, we utilized the following methods, each with its unique characteristics and requirements. 1). Methods without ground truth 3D label supervision: DFNet [10] and Direct-PoseNet [11] is the absolute pose regression method based on NeRF; FeatLoc++ [2] is the absolute pose regression method using sparse features. Furthermore, featLoc++Au utilizes sparse 3D structure to enhance visual localization performance; TransPoseNet [33] achieve visual localization based on attention mechanism; 2). Methods with ground truth 3D label supervision: SA [5] is a recently proposed absolute pose regression method based on 3D structure. HACNet [21], DSAC* [8] and DSAC++ [7] are state-of-the-art visual localization methods based on coordinate regression. PixLoc [30] is an advanced structure-based visual localization method.

**Implementation Details.** Our approach utilizes Nerfacto [38], an accelerated variant of NeRF. Nerfacto was trained for 100,000 iterations, and all camera poses are only centralized, with no scale adjustment. As errors in the ground truth pose of 7-Scenes dataset impair the training quality of Nerfacto [38], we turn on the pose-optimization component of the Nerfacto for 7-Scenes dataset. We set the far plane to 8 meters for office5a and office5b in the 12-Scenes dataset and set the far plane to 6 meters for other scenes. We set the near plane to 0 meters for all scenes. The coordinate regression network is trained for 40 epochs with a learning rate of 0.001. We empirically set $\tau = 200$ for the threshold of coordinate distance. In pose optimization, we use SuperPoint with SuperGlue and their official weights to detect correspondences, set $\lambda = 0.5$, $N_1 = 100$ for IBVS iterations. For training time, the Nerfacto and coordinate regression network are trained on one NVIDIA RTX3090 GPU for about 2 days. Please see the supplementary material for more implementation details.

Table 3: Comparison of median position and orientation error $(m,°)$ on the 7-Scenes dataset.

|  | Method | Chess | Fire | Heads | Office | Pumpkin | Kitchen | Stairs | Average |
|---|---|---|---|---|---|---|---|---|---|
|  | PoseNet [19] | 0.32/8.12 | 0.47/14.4 | 0.29/12.0 | 0.48/7.68 | 0.47/8.42 | 0.59/8.64 | 0.47/13.8 | 0.44/10.4 |
|  | Direct-PoseNet[11] | 0.10/3.52 | 0.27/8.66 | 0.17/13.1 | 0.16/5.96 | 0.19/3.85 | 0.22/5.13 | 0.32/10.61 | 0.20/7.26 |
| w/o | TransPoseNet [33] | 0.08/5.68 | 0.24/10.6 | 0.13/12.7 | 0.17/6.34 | 0.17/5.6 | 0.19/6.75 | 0.30/7.02 | 0.18/7.81 |
| ground truth | FeatLoc++Au [2] | 0.07/3.66 | 0.17/5.95 | 0.10/7.57 | 0.16/5.20 | 0.11/3.86 | 0.20/6.43 | 0.16/8.57 | 0.14/5.89 |
| 3D labe | DFNet[10] | 0.05/1.88 | 0.17/6.45 | 0.06/3.63 | 0.08/2.48 | 0.10/2.78 | 0.22/5.45 | 0.16/3.29 | 0.12/3.71 |
|  | Ours | **0.03/1.18** | **0.03/1.17** | **0.02/1.12** | **0.04/1.39** | **0.08/2.36** | **0.07/2.21** | **0.08/2.07** | **0.05/1.64** |
| w/ | SA[5] | 0.08/2.17 | 0.21/6.14 | 0.13/7.93 | 0.11/2.65 | 0.14/3.34 | 0.12/2.75 | 0.29/6.88 | 0.15/4.55 |
| ground truth | DSAC++[7] | 0.02/0.5 | 0.02/0.9 | 0.01/0.8 | 0.03/0.7 | 0.04/1.1 | 0.04/1.1 | 0.09/2.6 | 0.04/1.1 |
| 3D label | DSAC*[8] | 0.02/1.10 | 0.02/1.24 | 0.01/1.82 | 0.03/1.15 | 0.04/1.34 | 0.04/1.68 | 0.03/1.16 | 0.03/1.36 |
|  | HACNet[21] | **0.02/0.7** | 0.02/0.9 | 0.01/0.9 | **0.03/0.8** | **0.04/1.0** | 0.04/1.2 | **0.03/0.8** | **0.03/0.9** |
|  | PixLoc[30] | 0.02/0.80 | **0.02/0.73** | **0.01/0.82** | 0.03/0.82 | 0.04/1.21 | **0.03/1.20** | 0.05/1.30 | 0.03/0.98 |

**Comparisons.** We report the median translation (cm) and rotation (°) errors [19] in each scene, which is a main evaluation metric for localization accuracy. The localization accuracy is reported in Table 3 for 7-Scenes and Table 4 for 12-Scenes. As can be observed, our method consistently achieves the best performance among visual localization methods without ground truth 3D label

Table 4: Comparison of median position and orientation error $(m,°)$ on the 12-Scenes dataset.

| | w/o ground truth 3D label | | | | w/ ground truth 3D label | |
| --- | --- | --- | --- | --- | --- | --- |
| Scene | PoseNet[19] | FeatLoc++[2] | FeatLoc++Au[2] | Ours | SA[5] | HACNet[21] |
| Kitchen1 | 0.29/15.48 | 0.53/14.0 | 0.32/5.19 | **0.01/0.60** | 0.08/4.45 | **0.01/0.4** |
| Living1 | 0.29/15.31 | 0.55/9.75 | 0.26/3.89 | **0.01/0.57** | 0.08/2.50 | **0.01/0.4** |
| Bed | 0.57/17.85 | 0.60/10.2 | 0.37/5.39 | **0.02/0.82** | 0.07/3.87 | **0.01/0.4** |
| Kitchen2 | 0.21/18.18 | 1.21/29.1 | 0.73/6.37 | **0.02/0.79** | 0.08/2.96 | **0.01/0.3** |
| Living2 | 0.31/23.58 | 0.73/11.4 | 0.40/5.71 | **0.03/1.07** | 0.09/3.13 | **0.01/0.4** |
| Luke | 0.35/20.07 | 0.59/11.4 | 0.33/4.85 | **0.03/1.20** | 0.12/4.82 | **0.01/0.5** |
| Gates362 | 0.27/16.71 | 1.03/8.95 | 0.52/5.22 | **0.02/0.75** | 0.06/2.04 | **0.01/0.4** |
| Gates381 | 0.37/20.52 | 0.73/12.9 | 0.42/6.23 | **0.02/0.82** | 0.12/5.02 | **0.01/0.6** |
| Lounge | 0.29/18.42 | 0.97/8.30 | 0.39/4.50 | **0.02/0.83** | 0.07/2.53 | **0.01/0.5** |
| Manolis | 0.22/17.45 | 0.66/11.9 | 0.30/4.67 | **0.01/0.62** | 0.09/3.52 | **0.01/0.5** |
| Office5a | 0.57/14.55 | 0.61/8.05 | 0.31/4.32 | **0.06/2.63** | 0.10/5.08 | **0.01/0.5** |
| Office5b | 0.47/15.49 | 0.51/7.29 | 0.23/4.14 | **0.02/0.74** | 0.09/2.57 | **0.02/0.5** |
| Average | 0.35/17.80 | 0.73/11.9 | 0.38/5.04 | **0.02/0.95** | 0.09/3.54 | **0.01/0.5** |

supervision. Meanwhile, our method has comparable performance with methods using ground truth 3D labels as supervision, which closes the gap between the two types of visual localization methods. It is important to note that we use fewer posed images for training compared to the baseline methods, which shows the high value in practical application. Compared with the methods using NeRF for data augmentation, our method (0.05m, 1.64°) outperforms DFNet [10] (0.12m, 3.71°) and Direct-PoseNet [11] (0.20m, 7.26°), thus providing a more efficient way to take advantage of the scenes prior provided by NeRF. Moreover, our method outperforms attention-based MS-Transformer [27] and FeatLoc++Au [2], which utilize sparse 3D structure augmentation, highlighting the potential of NeRF in visual localization.

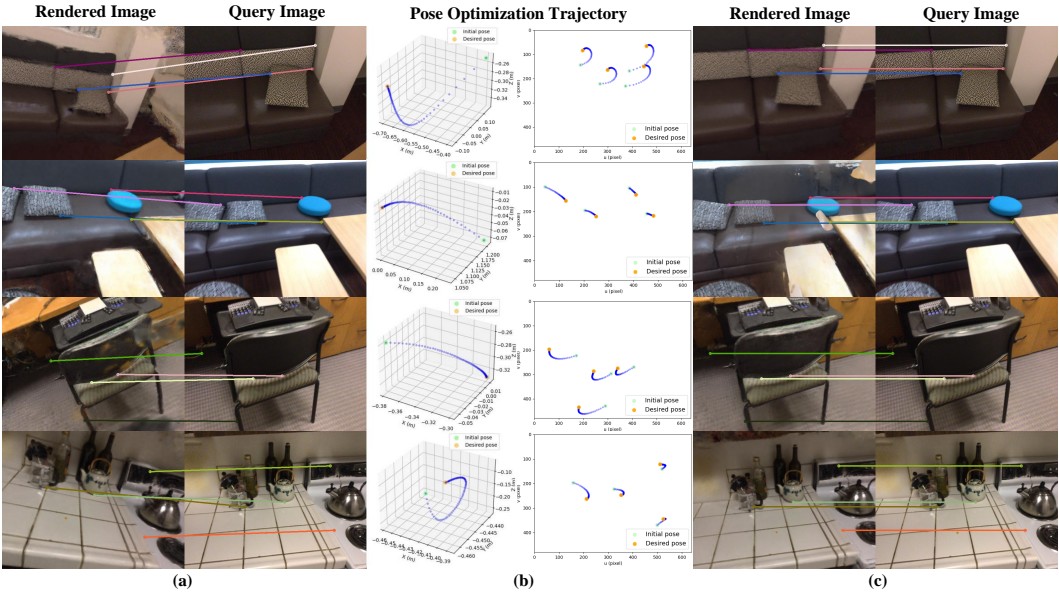

Figure 3: Qualitative results on 12-Scenes dataset. (a) the correspondences between images rendered using coarse poses and query images. (b) the pose optimization trajectories in 3D space and image plane. (c) the correspondences between images rendered using optimized poses and query images.

**Qualitative results.** We also qualitatively analyze the NeRF-IBVS on 12-Scenes dataset. Our coarse pose estimation can obtain the poses near the target as shown in Figure 3(a), which provides a good starting point for pose optimization. The pose optimization trajectories in both 3D space and image plane show that the optimized pose can accurately reach the desired pose as shown in Figure 3(b). Figure 3(c) shows that the images rendered using the optimized poses are accurately aligned with the

query images. Figure 3(b)-(c) demonstrate that our method can achieve accurate localization. Please see the supplementary material for more qualitative results.

## 5.2 Simulation for Navigation

To verify the effectiveness of our method in enabling IBVS-based navigation without using custom markers and the depth sensor, we randomly selected a pair of images from the real 12-Scenes dataset with co-views to serve as the initial state and the desired state for navigation. We first establish the correspondences between the initial state and the desired state as shown in Figure 4(a). Appropriate 2D and 3D correspondences are selected as navigation prior as shown in Figure 4(b)-(c). A 6-degree-of-freedom navigation simulation is then performed using the camera model. Figure 4(d) shows that the navigation trajectory in both 3D space and image plane can accurately reach the desired state. Figure 4(e) shows that the coordinate error of correspondences can converge. The coordinate error of correspondences can reflect errors in the pose. The simulation results in Figure 4(d)-(e) demonstrate that our IBVS-based navigation without using custom markers and the depth sensor can achieve accurate navigation. Please see the supplementary material for more simulation results.

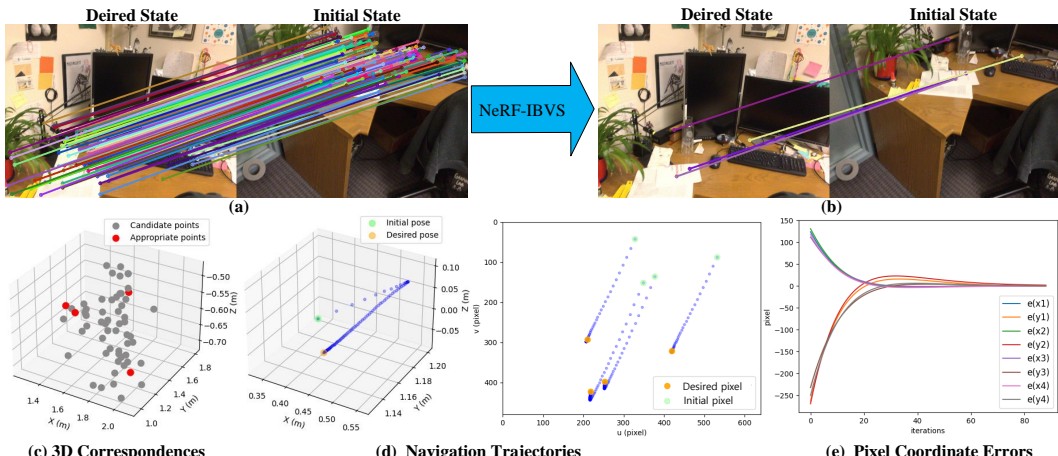

Figure 4: Simulation navigation results on 12-Scenes dataset. (a) candidate correspondences are established between initial states and desired states. (b) and (c) the appropriate 2D and 3D correspondences are selected by NeRF-IBVS. (d) navigation trajectory in both 3D space and image plane. (e) $e(x_i), e(y_i)$ is the image coordinate error of correspondences.

## 5.3 Ablation Studies

In this section, we conduct ablation experiments on the 12-Scenes dataset to evaluate the effectiveness of the major design of the proposed NeRF-IBVS.

**Effectiveness of coarse pose estimation.** To guarantee excellent performance for subsequent pose optimization, coarse pose estimation is required to output coarse poses and coarse 3D coordinates near the target. Therefore, we evaluate the localization performance of coarse pose estimation on the 12-Scenes dataset. As shown in Table 6, our coarse pose estimation obtains poses and 3D coordinates near the target (0.27m, 13.10°), providing a good starting point for pose optimization.

We consider that the coarse poses and coarse 3D coordinates near the target when the position and orientation error of the coarse pose reduces after pose optimization. To further verify the effectiveness of coarse pose estimation, we count how often this happens. The specific quantitative results on the 12-Scenes dataset as shown in Table 5, the average frequency of coarse pose error reduction is greater than 95%. Therefore, the coarse pose estimation consistently outputs results near the target.

**Effectiveness of pose optimization.** To verify the effectiveness of pose optimization, we compare the overall NeRF-IBVS with a coarse pose estimation pipeline on 12-Scenes dataset. Localization result in Table 6 shows that pose optimization (0.02m, 0.95°) can significantly improve the localization performance compared to coarse pose estimation (0.27m, 13.10°).

Table 5: Frequency of the coarse pose reduction after pose optimization on the 12-Scenes dataset.

| 12Scenes | Kitchen1 | Living1 | Bed | Kitchen2 | Living2 | Luke | Gates362 | Gates381 | Lounge | Manolis | Office5a | Office5b | Average |
|---|---|---|---|---|---|---|---|---|---|---|---|---|---|
| Frequency | 96.88% | 99.59% | 94.61% | 96.67% | 95.42% | 97.76% | 100.00% | 95.16% | 99.69% | 90.46% | 90.14% | 91.60% | 95.66% |

Table 6: Average of median position and orientation error $(m, °)$ on the 12-Scenes dataset.

| | coarse pose estimation | NeRF-IBVS w/0 correspondence selection | NeRF-IBVS w/ correspondence selection |
|---|---|---|---|
| 12-Scenes | 0.27/13.10 | 0.09/3.73 | **0.02/0.95** |

**Effectiveness of correspondence selection.** To verify the effectiveness of correspondence selection, we evaluate the localization performance of NeRF-IBVS without correspondence selection. The results presented in Table 6 show that the NeRF-IBVS without correspondence selection downgrade localization performance (0.09m, 3.73°) compared to NeRF-IBVS with correspondence selection (0.02m, 0.95°). It demonstrates that correspondence selection can select the appropriate correspondences for NeRF-IBVS and enhance the robustness of NeRF-IBVS to noise.

We also qualitatively analyze the effectiveness of correspondence selection. Coordinate error and depth error of correspondences are generally prone to occur in areas with poor rendering quality. As shown in Figure 5, the correspondence selection tends to choose correspondences that occur in areas with good rendering quality, which can enhance the robustness of pose estimation.

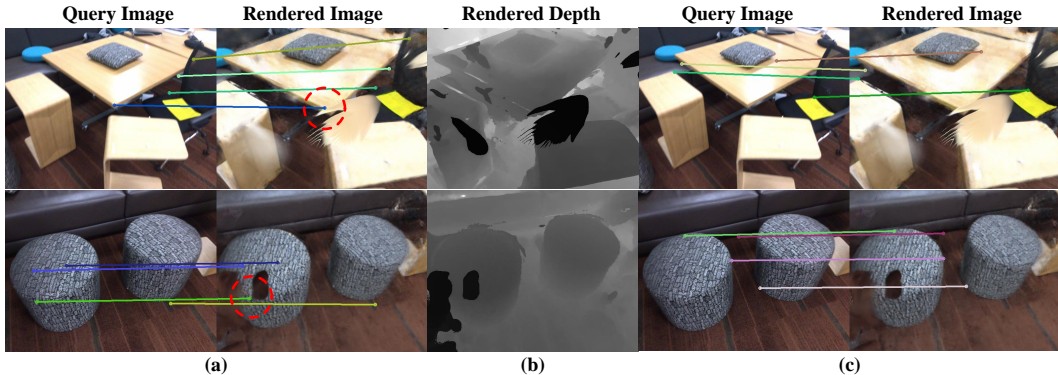

Figure 5: Qualitative comparison on correspondence selection. (a) the correspondences are randomly selected. The dashed red circles indicate the correspondences in the areas with poor rendering quality. (b) the rendered depth is used to indicate depth error of correspondences. (c) the correspondences are selected using correspondence selection algorithm.

## 6 Conclusion

We propose a novel localization framework that can achieve precise localization performance using only a few posed images compared to other localization methods. We first use a few posed images with coarse 3D labels provided by NeRF to train a coordinate regression network. The coarse pose of the query image can be estimated by a coordinate regression network with the PNP algorithm. Then, we optimize the coarse pose using IBVS based on NeRF representation to achieve accurate localization performance. Further, our method can provide effective navigation prior, which can enable IBVS-based navigation without using custom markers and the depth sensor. This advantage greatly expands the application scope of IBVS-based navigation. To the best of our knowledge, NeRF-IBVS is the first method that can accomplish both visual localization and navigation based on IBVS, providing a novel idea for pose optimization by controlling camera motion.

**Limitations.** The quality of NeRF rendering is crucial for our method. Due to the poor rendering quality of NeRF in large outdoor scenes compared to indoor scenes, our method is mainly applied in indoor scenes. Moreover, due to the slow rendering speed of NeRF, our method cannot achieve real-time performance. The proposed method also struggles with non-Lambertian and dynamic objects since the NeRF performs poorly in these scenarios. In the future, the improved NeRF variants can be used to improve the performance of our method.

## Acknowledgments

This work is supported by the National Natural Science Foundation of China (No. 91948303), Shanghai Municipal Science and Technology Major Project (2021SHZDZX0102), and CCF-Alibaba Innovative Research Fund For Young Scholars.

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
