# OpenReview forum: "NeRF-IBVS: Visual Servo Based on NeRF for Visual Localization and Navigation"
_NeurIPS.cc/2023/Conference — NeurIPS 2023 poster_

### Official Review · Reviewer_F153 · 2023-06-25

**Soundness:** 2 fair
**Presentation:** 2 fair
**Contribution:** 3 good
**Rating:** 5
**Confidence:** 4

**Summary:**

This paper addresses visual localisation (i.e. the estimation of the
6-degrees-of-freedom pose of a camera) from scene coordinate regression.

The problem tackled is the need for a large amount of pose and depth labels to
train scene coordinate regression networks.

The paper proposes a solution to reduce the label requirements by producing
pseudo-groundtruth depth labels with a NeRF trained on the scene.

The lower data requirement comes at the cost of a lower accuracy of the scene
coordinate regression network. This is compensated by a pose refinement step
based on visual servoing: the coarse pose is refined until the view rendered by
the nerf on the estimated pose matches the target pose. More specifically, the
pose is estimated until the local features in the rendered view are close to
their corresponding points in the target image. The pose updates are estimated
to reduce the discrepency between these feature positions.

The method proceeds as follows:
- Given a set of calibrated images of the scene, train a nerf.
- Generate pseudo-groundtruth depth with the nerf.
- Train the scene coordinate regression with the pseudo-groundtruth depth.
- Infer a coarse pose from the trained scene coordinate regression network.
- Refine the coarse pose using visual-based servoing

The method is compared against state-of-the-art localisation method from
several categories: scene coordinate regression network based (e.g. DSAC++),
pose estimation network based (e.g. PoseNet, Direct-PoseNet,  DFNet) and
feature based (PixelLoc).

**Strengths:**

S1: The problem of reducing the data requirement to train scene coordinate
regression network is an interesting problem.

S2: The approach to use NeRF to produce pseudo-groundtruth 3D labels and the use of
this synthetic depth inside a localisation experiment is insightful.

S3: The paper cites and compares against relevant works.

**Weaknesses:**

W1: The writing in general is good and one get the high-level description of the
technical derivations. However, some parts are confusing which requires the
reader to re-read a paragraph multiple times or to guess the technical
derviation being described.  (See Limitations for detailed comments and
suggested updates). Also, there are several english typos that can easily be
caught with an automatic spell-grammar checker.

W4: The paper claims that one advantage of the method is that it requires less
labeled data (calibrated images with depth) than other methods but there is no
experiments to support this claim (e.g. gradually reducing the number of
training images and comparing the performance drop).

**Questions:**

Q1:  L146,147: what is the difference between $m_r$ and $n_r$? from reading, one
  gets that $m_r$ is a pixel.

Q2: L171: does the next ibvs run starting from the output pose of the previous
  ibvs? Or do all the ibvs start from the coarse pose from the scene coordinate
  regression network?

Q3:  L187: where is $\hat{n}^r$ used?

Q4: L188: where is $d$ used?


**Limitations:**

Figure 1 would gain from having a technical description in addition to the
current caption. Something along the lines of: "Left: The coarse pose estimated with
the scene coordinate regression network is refined with IBVS. Right: Illustration of the pose updates / navigation in the NerF scene."

Technical typos:
- L4: "require groundtruth 3D labels for supervision": only regression-based
  methods require 3D labels (i.e. depth). Structure-based appraoches do not.
- L39: the reference [25] indeed uses nerf for data augmentation but [10,11]
  uses nerf to refine camera poses with a render-and-compare strategy (10 with
  photometric loss and 11 with a feature loss). This sentences need to be
  updated accordingly.
- Unsuported claim that the proposed method need less training images:
  - L46: "which is significantly fewer than typical visual localization method".
    This claim is not supported.
  - L86-87: "Our method ..."
  - L103: "We utilize few posed images"
  - L241: "It is important to note that we use fewer ..."
- The paper claims that RNR-Map [19] requires RGBD but the original paper
  claims that it is based on RGB vision only. It is would be useful to have
  this clarified.
- Eq1: M is undefined even though one can guess that it is the number of points
  along the ray
- L116: $t_n$ is undefined
- Eq2: Isn't the depth $D$ already defined in the camera coordinate frame? If
  so, then there is no need to transform the points with $T^{-1}$ i.e. it
  should be $P_W = D K^{-1} p$.
- L146: "depth of correspondences" -> "depth of pixels"
- Eq7 assumes that the set of correspondences remain the same for all IBVS,
  which is an assumption only introduced later in 3.2.2. Instead, the
  assumption should be mentionned before, even if only in a short sentence. The
  reader can then be referred to 3.2.2 for more details.

- L201: "we can obtain accurate ...": how are these correspondences derived?
  one can guess that is with L144-147 but it would be helpful to the reader if
  it was explicitly mentionned.

- L226: Pixloc is actually a feature-based refinement only method so it does
  not fully qualify as a structure-based method.

- Fig4: e(x) ... are undefined even though one can guess that these are
  distance errors



Confusing writing that impedes with the understanding of the technical details:
- L7: "only a few". It might be better to either give a quantification of how
  much less or only write "less" images.

- L8: "To achieve this, we first use a few posed images with coarse 3D
  labels provided by NeRF to train a coordinate regression network, which is
  used to provide the coarse pose for unseen view. "
  It would be easier to read if written with a different order:
  - pseudo ground-truth 3d labels are computed with nerf
  - the scene coordinate regression network is trained
  - a coarse pose is estimated from the regression network
  - the coarse pose is refined with IBVS

- L53: "the correspondences with depth": the reader can guess the derivation
  bein run i.e. select pixels in one image and find their match in the second
  image using the depth and the image poses. But the reader should not have to
  guess the technical derivations so it is better to write it down explicitly.
  - Same comment for L141: the "depth correspondences"

- L61: "navigation prior" is undefined. Does this refer to the "simulated"
  navigation run in the NeRF scene?

- L125: "geometric constraints" is undefined

- L127: Does "edge areas" refer to the scene's edge or the image's edge? One
  could guess that it is the scene's edge but it is better to specify it.

- L136:
  - "we first initialize IBVS ...". At this stage, IBVS has not been
    introduced and it is not part of the common knowledge in the visual
    localisation community (this is more of a robotic common knowledge). So it
    would be helpful to the reader to have a high level description of IBVS
    before the low-level one that is already in the paper. This will help the
    reader understand what "initializing the IBVS" means (e.g. "The initial pose
    in the IBVS optimisation is the coarse pose and the 3D points that will
    support the optimisation are generated with the NeRF depth.").
  - the term pixel velocity is not introduced and it is not a term common in
    the visual localisation community which is more used to the
    render-and-compare notations than the visual servoing notations.

- L168: this sentence is paradoxial. An alternative formulation could be: "The
  assumption is violated so accumulation errors will occur."


A non exhaustive list of typos:
- L26: the other line of approach *is* [...] regresse*s*
- L75: "To achieve ...". It seems this sentence should not be there.
- L76: has attracted
- L103: "Then the same ...": this sentence misses a verb
- Fig4: Deired -> Desired


### Post-Rebuttal

#### Addressed weaknesses:
W3: The evaluation on 12 scenes compares only against methods that do not require
groundtruth depth labels, whereas the 7 scenes evaluation also compared against
method that do use groundtruth depth labels.
- Update: The paper does evaluate methods using ground-truth depth (only DSAC is missing in the 12 scenes experiments).

W2: The paper claims to contribute not only to localisation but also navigation but
there is no comparison to previous navigation work (RNR-MAP). Also, the claim
that RNR-MAP requires depth information seems opposite to the RNR-MAP's claim
to be RGB-vision based only (in the related work section).
- Update: the comparison to rnr-map is not applicable so this limitation is obsolete.
However, the contribution to the navigation task should be clarified.

---

> ### Author Rebuttal · Authors · 2023-08-09
>
> Thank you for the constructive comments and suggestions.
>
> **W1: Paper writing could be improved.**
>
> Thanks for the valuable suggestion. We have polished the whole paper to make it more smooth and concise for readers. Please refer to **Q2** in the **"General Response"** for specific modification details.
>
> **W2: The paper claims to contribute not only to localization but also navigation but there is no comparison to previous navigation work (RNR-MAP). Also, the claim that RNR-MAP requires depth information seems opposite to the RNR-MAP's claim to be RGB-vision based only.**
>
> Thanks for the valuable suggestion.
> 1) RNR-MAP [1] indeed uses depth maps in the construction of the map, which is described in the section “3. RNR-Map” of RNR-MAP. Also, the inputs of the navigation system use depth information, which is described in "Figure 3. Navigation System Overview".
> 2) Our IBVS-based navigation (6-degree-of-freedom navigation, without visual odometry information) is not configured in the same way as RNR-MAP (3-degree-of-freedom navigation, with visual odometry information). Therefore, the comparison between IBVS-based navigation and RNR-MAP is unfair.
> 3) Our main contribution to visual navigation is to enhance IBVS-based navigation. The main point is that our method enables navigation based on IBVS without using custom markers and the depth sensor compared to general IBVS-based navigation methods [9,17]. It is obvious to enhance the general IBVS-based navigation. Therefore, it does not need to be compared with other visual navigation methods to illustrate enhancement.
>
> **W3: The evaluation on 12 scenes compares only against methods that do not require groundtruth depth labels, whereas the 7 scenes evaluation also compared against method that do use groundtruth depth labels.**
>
> Thanks for your valuable comment. Actually, our method is compared with both types of methods in 7Scenes and 12Scenes.
> Specifically, in the 7Scenes dataset, our method is compared to both methods that without using groundtruth depth labels (DFNet [10], FeatLoc++Au [2], MS-Transformer [26, TransPoseNet [32], PoseNet [18]) and methods that use groundtruth depth labels (PixLoc [29], HACNet [20], DSAC* [8], DSAC++ [7], SA [5]).
> In the 12Scenes dataset, our method is compared to both methods that without using groundtruth depth labels (FeatLoc++Au and FeatLoc++ [2], PoseNet [18]) and methods that use groundtruth depth labels (SA [5], HACNet [20]).
>
> **W4: There are no experiments to support this claim that one advantage of the method is that it requires less data than other methods.**
>
> Thanks for your valuable comment.
> Please refer to **Q1** in **“General Response”** for the number of training data. Actually, we have included these tables in the section **"1.1 Amount of Training Data for Each Scene"** of the supplementary material. And we will include these tables in the paper to make it clear to readers.
>
> **Q1: L146,147: what is the difference between $m_r$ and $n_r$ from reading, one gets that $m_r$ is a pixel.**
>
> $n_r$ is defined in L147 as the image coordinate of the correspondences. Specifically, $n_r$ is the pixel coordinate $m_r$ minus the pixel coordinate of the camera's optical center. We will add the explanation of $n_r$ to the paper.
>
> **Q2: L171: does the next ibvs run starting from the output pose of the previous ibvs?**
>
> The next ibvs run starts from the output pose of the previous ibvs and we will add this detail to the paper.
>
> **Q3: L187: where is $\mathbf{\hat{n}_r}$ used?.**
>
> In L188, $\mathbf{\hat{n}_r}$ is used to compute the coordinate distance $d = ||\mathbf{\hat{n}_r} - \mathbf{n_r}||_2$
>
> **Q4: L188: where is $d$ used?**
>
> In "3.2.2 Correspondence Selection", we empirically set a threshold $\tau$ for the coordinate distance $d$. We will filter out the outliers in the correspondences when the coordinate distance $d$ is greater than the threshold $\tau$. We will add the symbol $d$ after the coordinate distance in L191 to make it clear for readers.
>
> **Limitations: Technical typos**
>
> Thanks for the valuable suggestion, we have carefully studied all the comments. Below we will provide point-by-point responses to all the suggestions.
> 1) We update "while state-of-the-art regression-based methods require dense ground truth 3D labels for supervision" in L4.
> 2) We update "Current NeRF-based visual localization methods employ NeRF for data augmentation purposes [25], or use nerf to refine camera poses with a render-and-compare strategy [10,11]" in L39.
> 3) Please refer to **W4**.
> 4) Please refer to **"1."** in **W2**.
> 5) We add "M is the number of points along the ray" for Eq1.
> 6) We add "$t_n$ is the near bound of the camera ray" in L116.
> 7) Depth $D$ is defined in the camera coordinate frame. But $D\mathbf{K}^{-1}\mathbf{p}$ only transforms points from the image coordinates frame to the camera coordinates frame. We require camera pose $\mathbf{T}$ to transform the point from the camera coordinate frame to the world coordinate frame: $P_w=D\mathbf{T}^{-1}\mathbf{K}^{-1}\mathbf{p}$  in Eq2.
> 8) In L146, the depth of the correspondences is used to initialize the Jacobi matrix of IBVS. Therefore, depth $Z_c$ is the depth of the correspondences.
> 9) We add "The set of correspondences remains the same for all IBVS" in the "3.2.1 IBVS Module" for Eq7.
> 10) In L136, we have rewritten the method section to make IBVS module clearer for readers.
> Please refer to the beginning of **Q2** in the **"General Response"** for specific modification details.
> We add "In IBVS, the pixel velocity indicates the desired velocity to control the correspondences motion towards the target coordinate which is the image coordinates error of correspondences multiplied by a scaling factor".
> 11) For issues in L201, Fig4, L7, L8, L53, L61, L125, L127, and L168, please refer to **Q2** in the **"General Response"** for specific modification details.
> 12) For the typos. We perform grammatical checks on the whole paper and correct all grammatical errors.

---

> > ### Comment · Reviewer_F153 · 2023-08-14
> > **Thank you for the informative rebuttal**
> >
> > The rebuttal addresses all the comments and questions raised in the review, this is very much appreciated.
> > The polishing of the writing is indeed a very important part of the paper value so the additional effort is well valuable.
> >
> > W3: Indeed, the paper evaluates methods that use ground-truth depth on 12 scenes. The review will be updated accordingly.
> > However, the DSAC family is not reported when it is considered one of the state-of-the-art method. This is a bit surprising especially given that the authors do report the DSAC results on 7-scenes.
> >
> > W2: The previous comment on RNR-map being RGB-based is due to a reference confusion: RNR-map is described in [19] and not [1]: Vision-Only Robot Navigation in a Neural Radiance World, with [1] indeed running rgb-based only navigation. The review will be updated accordingly.
> >
> > #### Additional comments:
> > - Section 3.2.2. describes the filtering of the pixel correspondences established in 3.2.1 so a more reader-friendly title could be "Correspondence filtering" or "Correspondence Geometric Verification".
> >
> > #### Additional questions:
> > - Could the contribution related to the visual navigation be clarified? The current understanding is that the IBVS-Nerf is helpful to gather pixel correspondences without the need for any external markers. These pixel correspondences will then be used in the IBVS.
> > - "Navigation" prior is not a standard term. Could it be specified? (e.g. is it a set of poses the navigation will go through? is it the initial position the navigation will start from?)

---

> > > ### Author Response · Authors · 2023-08-14
> > > **Response to additional comments of Reviewer F153.**
> > >
> > > We sincerely thank you for your efforts in reviewing our paper and your constructive suggestions again. We really enjoy communicating with you.
> > >
> > > **W2**: Sorry for the reference confusion.  We update the references “Nerf-nav [1] designs a smooth and collision-proof navigation strategy based on the density provided by NeRF” in “Related Work”.
> > >
> > >
> > > **W3**: DSAC++ [7] and DSAC*  [8] do not report specific median errors in the 12Scenes dataset. Furthermore, HACNet  [20] has better performance than DSAC* which is the most advanced method in the DSAC family, based on the evaluation of indoor 7Scenes (**0.03/0.9** vs 0.03/1.36) and outdoor Cambridge (**0.3/0.16** vs 0.34/20.6). Therefore, we consider that the comparison between HACNet and our method is sufficient.
> > >
> > >
> > >
> > > **Additional comments**
> > >
> > > Thanks for the valuable comments, we update the title of "Section 3.2.2" to "Correspondence filtering".
> > >
> > > **Additional questions**
> > >
> > > Thanks for the valuable questions.
> > > 1. "Navigation prior" is non-collinear correspondences and navigation trajectories corresponding to the correspondences, where
> > > the navigation trajectories are the set of poses the navigation will go through. We add the details in the paper to make it clear to readers.
> > >
> > > 2. General IBVS-based navigation [9,17] requires an external maker to obtain correspondences that meet the requirement of IBVS-based navigation. Specifically, IBVS-based navigation requires correspondences that are non-collinear and remain in the camera’s field of view during navigation. Based on the navigation prior provided by NeRF-IBVS, the correspondences that meet the requirement of IBVS-based navigation can be obtained without the external maker. Also, general IBVS-based navigation requires the depth sensor to get the depth of correspondences to update the Jacobi matrix of IBVS. However, our enhanced IBVS-based navigation does not require a depth sensor. Specifically, NeRF-IBVS can back-project the correspondences to get 3D points (3D correspondences) based on the rendered depth of NeRF at the beginning of navigation.  During navigation, the depth of the correspondences is obtained by projecting the 3D points using the current pose to update the Jacobi matrix of IBVS. In summary, our method enables IBVS-based navigation without using external markers and the depth sensor compared to general IBVS-based navigation methods.
> > > Please refer to **"3.3 Visual Navigation"** for the details and we will further clarify the contribution related to the visual navigation in the paper.

---

### Official Review · Reviewer_91GX · 2023-06-27

**Soundness:** 3 good
**Presentation:** 2 fair
**Contribution:** 2 fair
**Rating:** 4
**Confidence:** 3

**Summary:**

The paper presents a novel visual localization method combined with the NeRF technique to address the issue of training a large number of pose images in existing visual localization models. The proposed method has two advantages: (1) the paper trains a coordinate regression network using a few posed images with coarse 3D labels generated by NeRF. (2) The paper uses an image-based visual servo to explore 3D scenes for pose optimization.  The simulation experiments show superior performance in comparison with other methods.

**Strengths:**

The proposed method, NeFT-IBVS uses a coarse-to-fine strategy to accurately localize the camera with a few posed images being adopted.  The method equips with two leading edges: (1) a NeRF-based posed images are used to train a coordinate regression network in the coarse stage. (2) an optimization algorithm is presented to effectively optimize the coarse pose.  Furthermore, the authors also design a new pipeline to reduce the rendering frequency of NeRF to speed up the pose optimization.

**Weaknesses:**

1. The authors claim the proposed method is able to localize the camera with fewer posed images, can the authors give a rigorous description of it? For instance, how many posed images can be used, or can the authors show the reduced ratio of used posed images compared to state-of-the-art methods?
2. Can the authors give a comparison with DSAC method[1] on the 7-Scene dataset? It seems that the proposed method performs worse than it.
[1] Brachmann, Eric and Carsten Rother. “Visual Camera Re-Localization From RGB and RGB-D Images Using DSAC.” IEEE Transactions on Pattern Analysis and Machine Intelligence 44 (2020): 5847-5865.

3. For statistical significance, the experiments should be conducted several times and the statistical significance of the results should be determined. Furthermore, some typos are necessary to be corrected. For example, “We” should be rewritten “we” in Line 103. “enables” should be rewritten “enable”. The writing is obligated to be further polished.


**Questions:**

See Weakness part

**Limitations:**

Due to the poor rendering quality of NeRF in large outdoor scenes compared to indoor scenes, our method is mainly applied in indoor scenes. Moreover, due to the slow rendering speed of NeRF, the proposed method cannot achieve real-time performance.

---

> ### Author Rebuttal · Authors · 2023-08-09
>
> Thank you for the constructive comments and suggestions.
>
> **W1: The authors claim the proposed method is able to localize the camera with fewer posed images, can the authors give a rigorous description of it? For instance, how many posed images can be used, or can the authors show the reduced ratio of used posed images compared to state-of-the-art methods?**
>
> Thanks for your valuable suggestion.
> Please refer to **Q1** in **“General Response”** for the number of training data. Actually, we have included these tables in the section **"1.1 Amount of Training Data for Each Scene"** of the supplementary material. And we will include these tables in the paper to make it clear to readers.
>
>
> **W2: Can the authors give a comparison with DSAC method[1] on the 7-Scene dataset? It seems that the proposed method performs worse than it. [1] Brachmann, Eric and Carsten Rother. “Visual Camera Re-Localization From RGB and RGB-D Images Using DSAC.” IEEE Transactions on Pattern Analysis and Machine Intelligence 44 (2020): 5847-5865.**
>
> Thanks for your valuable suggestion.
> Actually, We have performed the comparison in the 7-Scene dataset and the results are given in Table 1 of the submitted manuscript. The mentioned method DSAC is denoted as DSAC* [8]. Although the localization performance of our method is slightly lower than DSAC*, we use fewer data to train the model **(2090 vs 26000)** and achieved comparable performance **(0.05m/1.55$^{\circ}$ vs 0.03m/1.36$^{\circ}$)**. Furthermore, our method can enhance IBVS-based navigation, which enables navigation based on IBVS without using custom markers and the depth sensor. We have included references to baseline methods in all tables to make it more clear for readers.
>
> **W3: For statistical significance, the experiments should be conducted several times and the statistical significance of the results should be determined. Furthermore, some typos are necessary to be corrected. For example, “We” should be rewritten “we” in Line 103. “enables” should be rewritten “enable”. The writing is obligated to be further polished.**
>
> Thanks for your valuable suggestion.
> 1) We conduct five experiments on the 7Scenes and 12Scenes datasets and calculate the mean and variance of the results, which are shown in the following table:
>
> |              | 7Scenes               | 12Scenes              |
> |--------------|-----------------------|-----------------------|
> | **Mean**     | 0.05m/1.61$^{\circ}$  | 0.02m/0.92$^{\circ}$  |
> | **variance** |   3.28$\times 10^{-6}$/0.00407                    |   6.45$\times 10^{-7}$/0.00079                     |
>
> As shown in the table, the mean is almost identical to the results presented in the paper and the variance is very small. Therefore the performance of the proposed method is very stable.
>
> 2) We have corrected “We” to “we” in Line 103.
> We have corrected “enables” to “enable” in Figure 1 and Line 62. In addition, we further polished the whole paper and performed grammatical checks. Please refer to **Q2** in the **"General Response"** for specific modification details.

---

> > ### Comment · Reviewer_91GX · 2023-08-11
> >
> > Thanks for your rebuttal.

---

> > > ### Author Response · Authors · 2023-08-14
> > > **Thanks for your response！**
> > >
> > > We sincerely thank you for your efforts in reviewing our paper again. We hope we have resolved all your concerns.

---

### Official Review · Reviewer_k2yC · 2023-07-04

**Soundness:** 4 excellent
**Presentation:** 3 good
**Contribution:** 3 good
**Rating:** 6
**Confidence:** 4

**Summary:**

This paper proposes a visual localization pipeline that uses NeRF and image-based visual servoing. The main contribution is that the proposed method uses much fewer images with pose annotations and no 3D ground-truth labels (only uses pseudo ground-truth provided by NeRF to train a coordinate regression network). The method first relies on the coordinate regression network to provide an initial coarse pose, which is then refined with image-based visual servoing.

**Strengths:**

The paper presents a creative approach towards the visual localization problem. I believe the utilization of NeRF to learn a coordinate regression model and as pose initialization is novel, considering that prior works used NeRFs mostly for data augmentation. Furthermore, formulating the pose refinement process as a visual servoing problem that uses state-of-the-art image matching methods and NeRF is an interesting idea which fits well the problem. Overall, the paper proposes something new and useful to an important task.

**Weaknesses:**

I have a few concerns/comments. Not all of these are necessarily weaknesses.

There is discussion over iNeRF in the related work. Authors claim that limitations of iNeRF are that it requires a very good initial pose and relies on photometric loss which is sensitive to artifacts in the rendered images. This is somewhat true for the proposed approach as well as visual servoing, by definition, requires visual overlap between observation and query image and any image matching method is going to be affected by rendering artifacts. In fact, the paper acknowledges this fact and has an extra verification step (section 3.2.2) to filter out outliers from unreliable Superpoint+Superglue correspondences.

In my understanding using NeRF to initialize IBVS is analogous to the structured methods (such as [27,36]) retrieving a set of candidate database images to establish an initial pose(s). The proposed method mentions that only renders a single image C_r, but how do you ensure that C_r has visual overlap with the query? How often does this occur? Why not sample around the coarse pose T and generate multiple images?

The paper uses the phrase "velocity s of the image coordinates". Isn't that just the optical flow?

In terms of pipeline complexity, the method is actually closer to the structured methods (than the regression-based). I am curious to see runtime required for a single query compared to the baseline methods.

Can you provide more details as to the number of posed images required to train the proposed method vs the baselines?

What is the difficulty of the queries in the visual servoing experiments? i.e. what is the pose difference / overlap between initial and desired state? It would be interesting to divide the test set into easy/hard queries based on their pose diff / ovelap in order to show the limitations of the method. The authors do show the performance of the coarse pose estimation but it is not clear how sensitive IBVS is to that coarse estimate.

Minor issue: Please include references of baselines in Table 1, and use "/" instead of "," to separate numbers.

**Questions:**

See weaknesses.

**Limitations:**

The authors mention a couple important limitations in the main paper. I could not find discussion on potential negative societal impact.

---

> ### Author Rebuttal · Authors · 2023-08-09
>
> Thank you for the constructive comments and suggestions.
>
> **W1: The proposed method mentions that only renders a single image $C_r$, but how do you ensure that $C_r$ has visual overlap with the query? How often does this occur? Why not sample around the coarse pose T and generate multiple images?**
>
> Thanks for your valuable suggestion.
> The coordinate regression network is able to ensure that the single rendered image $C_r$ corresponding to the estimated coarse pose overlaps with the query image. To verify this argument, we perform quantitative experiments on the 12Scenes dataset. We consider that the single rendered image corresponding to the coarse pose has a good visual overlap with the query image when the position and orientation error of the coarse pose reduces after pose optimization. Therefore, we count how often this happens, and the specific quantitative results are shown in the following table:
>
> | 12Scenes  | kitchen1 | living1 | bed    | kitchen2 | living2 | luke   | gates362 | gates381 | lounge | manolis | 5a     | 5b     | Average    |
> |-----------|----------|---------|--------|----------|---------|--------|----------|----------|--------|---------|--------|--------|--------|
> | **Frequency** | 96.88%   | 99.59%  | 94.61% | 96.67%   | 95.42%  | 97.76% | 100.00%  | 95.16%   | 99.69% | 90.46%  | 90.14% | 91.60% | 95.66% |
>
>
> As shown in the table, the visual overlap between rendered and query images occurs with high frequency.
>
> Since there is a high probability that the rendered image corresponding to the coarse pose has visual overlap with the query image and rendering multiple images spends a lot of time. Therefore, we do not sample around the coarse pose T and generate multiple images.
>
> **W2: The paper uses the phrase "velocity s of the image coordinates". Isn't that just the optical flow?**
>
> Indeed, both the velocity of the image coordinates in our paper and optical flow represent the velocity of pixels, but they are solved differently. In IBVS, the velocity of the image coordinates indicates the desired velocity to control the correspondences motion towards the target coordinate which is the image coordinates error of correspondences multiplied by a scaling factor. The optical flow is primarily solved based on the pixel intensity changes between consecutive frames.
>
> **W3: In terms of pipeline complexity, the method is actually closer to the structured methods (than the regression-based). I am curious to see the runtime required for a single query compared to the baseline methods.**
>
> Indeed, our method is more time-consuming due to the slow rendering speed of NeRF. For example, the time comparison of our method with state-of-the-art methods HACnet [20] in our device is (3.61 seconds vs 0.07 seconds). In the future, faster and more accurate variants of NeRF can be used to further improve the overall efficiency of the proposed method.
>
>
> **W4: Can you provide more details as to the number of posed images required to train the proposed method vs the baselines?**
>
> Please refer to **Q1** in **“General Response”** for the number of training data. Actually, we have included these tables in the section **"1.1 Amount of Training Data for Each Scene"** of the supplementary material. And we will include these tables in the paper to make it clear to readers.
>
>
> **W5: It would be interesting to divide the test set into easy/hard queries based on their pose diff / ovelap in order to show the limitations of the method. The authors do show the performance of the coarse pose estimation but it is not clear how sensitive IBVS is to that coarse estimate.**
>
> Thanks for your valuable suggestion.
> In order to illustrate the sensitivity of IBVS to coarse estimate, we conduct quantitative experiments on the 12Scenes dataset. There is no relevant literature in the field of IBVS to define easy and hard cases, we assume that the test image whose position and orientation errors of the coarse pose are greater than the median error is the hard case, and vice versa as the easy case.
> We consider that the IBVS module optimizes the pose successfully when the error of the coarse pose is reduced. We count the frequency of successful optimization of the coarse pose in the easy case and hard case respectively. Specifically, the quantitative experimental results are shown in the following table:
>
> |                  | kitchen1   | living1   | bed        | kitchen2   | living2    | luke       | gates362  | gates381   | lounge    | manolis    | 5a         | 5b         | Average        |
> |------------------|------------|-----------|------------|------------|------------|------------|-----------|------------|-----------|------------|------------|------------|------------|
> | **median error** | 0.23/13.41 | 0.19/9.65 | 0.38/15.26 | 0.28/14.79 | 0.34/14.32 | 0.27/11.52 | 0.11/8.31 | 0.23/13.12 | 0.39/9.78 | 0.32/15.60 | 0.27/16.24 | 0.25/15.21 | 0.27/13.10 |
> | **easy case**         | 100.00%    | 100.00%   | 99.02%     | 100.00%    | 100.00%    | 99.36%     | 100.00%   | 99.05%     | 100.00%   | 98.25%     | 99.20%     | 98.03%     | 99.41%     |
> | **hard case**         | 93.75%     | 99.19%    | 90.20%     | 93.33%     | 90.86%     | 96.15%     | 100.00%   | 91.27%     | 99.39%    | 82.71%     | 81.12%     | 85.22%     | 91.93%     |
>
> The results in the table show that IBVS can be run successfully almost all the time in the easy case and fails with a small probability in the hard case. Thus IBVS is robust to most of the coarse estimation of poses.
>
> **Minor issue: Please include references of baselines in Table 1, and use "/" instead of "," to separate numbers.**
>
> Thanks for pointing this out.
> We have added references to baselines in Table 1, and corrected "/" to "," to separate numbers in Table 1 and Table 2.

---

> > ### Comment · Reviewer_k2yC · 2023-08-13
> >
> > I would like to thank the authors for their answers to my comments. I appreciate both new quantitative experiments they provided in their rebuttal especially the results with respect to the robustness of IBVS to certain coarse pose error. Ideally, this experiment should have been carried out by manually choosing coarse poses (such that you can control the position and orientation error) that would allow a more systematic evaluation of the visual servoing. However, in the context of the overall system, the results do show robustness even with relatively large errors.
> >
> > The authors already mentioned they will include details on the amounts of training data, and I would like to re-iterate the importance of doing so in the main paper, since one of the main arguments for the proposed approach is that it uses less data.

---

> > > ### Author Response · Authors · 2023-08-13
> > > **Thank you for your insightful comments!**
> > >
> > > Thank you again for helping us make the paper stronger, we really enjoy communicating with you. We will include details on the amounts of training data in the main paper.

---

### Official Review · Reviewer_oADp · 2023-07-05

**Soundness:** 3 good
**Presentation:** 2 fair
**Contribution:** 2 fair
**Rating:** 5
**Confidence:** 4

**Summary:**

This paper tackles the task of visual localization, i.e. estimating a camera pose from a query image. Approaches in the literature require a large number of posed images and even dense 3D supervision for some of them. Authors propose to leverage Neural Radiance Fields (NeRF) to solve the problem of visual localization while requiring fewer posed images and no 3D labels.

The method can be decomposed into sequential steps: (1) A NeRF model is trained from a set of posed images, (2) a coordinate regression network is trained on the posed images using depth predictions from the NeRF model to provide coarse 3D labels, (3) a coarse camera pose is predicted by the coordinate regression network from the query image, and is refined by performing Image-Based Visual Servoing. Camera pose refinement is turned into a navigation problem from the coarse to the target pose.

The paper compares the localization performance of the proposed method with different baselines of the literature on 2 datasets: 7-scenes and 12-scenes. Authors show that their method outperforms reported baselines trained without 3D labels and is on par with the ones trained with dense 3D supervision while using less training posed images.

Finally, the authors show their NeRF-IBVS framework can also be used to perform camera navigation from an initial state to a target state, both specified as images.

**Strengths:**

1. The paper tackles an important problem: performing visual localization from fewer posed images (compared with current state-of-the-art methods) and no required 3D dense supervision.

2. NeRF models are an interesting tool to perform visual localization and it is great to have papers studying this application.

3. The paper considers a decent number of baselines from the literature in the experimental study. Ablation studies are also performed to evaluate the gain brought by the different proposed contributions.

**Weaknesses:**

1. [Major] Paper writing could be improved. The current version of the paper is fine when it comes to understanding the main contributions but several reads were needed to grasp the details of the work. I suggest authors go through the paper again and improve the coherence and writing.

2. [Major] NeRF is a differentiable function mapping and thus, as done in previous work (iNeRF [43] as mentioned by authors in the paper), pose refinement can be performed by freezing the NeRF weights and optimizing camera location based on a rendering loss. It would be interesting that authors compare using IBVS and optimization as in iNeRF to perform the last pose refinement step, i.e. from the coarse to the target position. Optimizing camera pose based on NeRF rendering can be considered as a simpler alternative to the IBVS framework (e.g. no need to build the $L$ matrix).

3. [Minor] This paper introduces a limited set of technical novelties. However, it tackles an important problem and nicely combines previous works, proposing simple yet efficient ideas to perform the target task. Thus, I consider this not an important issue with this work, but should still be mentioned in this review. For example, the authors mention they “design a new fast iteration strategy that reduces the rendering frequency of NeRF”. To the best of my knowledge, this is mainly about querying the NeRF model every n iterations, which might not be considered as a strong contribution. Authors might want to reconsider how they introduce this part of their work.

**Questions:**

1. [Major] Paper writing could be improved. Some sentences could be made smoother (e.g. some sentences are split in half while they should not). Writing could also be made more concise and efficient, as some parts are quite hard to follow. I am sorry to provide only vague directions here, but I have the feeling there is not one specific part of the paper to improve, but rather smoothness and conciseness should be improved globally to allow a better understanding. Technical sections (e.g. IBVS, correspondence selection) could be presented more progressively by defining some of the core notions early on and diving into the details more smoothly.

2. [Major] An experiment involving a comparison between IBVS and iNeRF-like camera optimization in the last step, i.e. refinement from coarse to target pose, should be performed by authors.

**Limitations:**

Authors mentioned two relevant limitations of their work:
1. NeRF rendering quality can be low in large outdoor scenes, which might have an impact on the visual localization performance. Experiments are only conducted in indoor scenes in this paper.
2. The low rendering speed of NeRF models prevents their solution from running in real time.

Both limitations rather come from the capabilities of current NeRF models than the proposed method. Future improvements in the field of NeRFs might allow better localization in large scenes and with faster runtime.

---

> ### Author Rebuttal · Authors · 2023-08-09
>
> Thank you for the constructive comments and suggestions.
>
> **W1 and Q1: Paper writing could be improved.**
>
> Thanks for your valuable suggestion. We have polished the whole paper to make it more smooth and concise for readers. Please refer to **Q2** in the **"General Response"** for specific modification details.
>
>
> **W2 and Q2: Optimizing camera pose based on NeRF rendering can be considered as a simpler alternative to the IBVS framework. An experiment involving a comparison between IBVS and iNeRF-like camera optimization in the last step.**
>
> Thanks for your valuable suggestion.
> 1) Since our other contribution is to enhance IBVS-based visual navigation. Therefore, our method will not accomplish the visual navigation task, if we replace IBVS framework with iNeRF.
> 2) Since iNeRF performs time-consuming neural rendering at each optimization iteration, iNeRF spends more time relative to IBVS framework (15.65 seconds vs 3.58 seconds). Therefore, replacing IBVS with iNeRF in our method will substantially increase the time cost of pose optimization.
> 3) Implementing the replacement of IBVS with iNeRF in our method and testing it on 7Scenes and 12Scenes datasets is not achievable in a short period of time. We think this replacement is a good direction for future research.
>
> **W3: “fast iteration strategy” might not be considered as a strong contribution.**
>
> Thanks for your valuable suggestion. The fast iteration strategy is simple but greatly accelerates the iteration of pose optimization. It provides a good starting point for possible future work related to IBVS utilizing the scene prior information provided by NeRF.

---

> > ### Comment · Reviewer_oADp · 2023-08-14
> >
> > I would like to thank the authors for their efforts in trying to address my concerns and the ones of other reviewers.
> >
> > 1. Writing was an important issue and I am happy authors spent time polishing the paper and adding details to allow an easier understanding of their method.
> >
> > 2. I still believe the comparison with iNeRF would be interesting, but I also understand this limited rebuttal time was too short for authors to conduct such a study. The provided running time comparison is interesting.
> >
> > 3. The additional table providing a comparison with baselines in terms of training data quantity (presented in the general response to all reviewers) is very valuable and should be added to the main paper as data efficiency is an important claim of the paper.

---

> > > ### Author Response · Authors · 2023-08-14
> > > **Thank you for your insightful comments!**
> > >
> > > We sincerely thank you for your efforts in reviewing our paper and your constructive suggestions again. We will include the table containing the number of training data in the main paper to demonstrate the data efficiency.

---

### Official Review · Reviewer_iS7u · 2023-07-10

**Soundness:** 3 good
**Presentation:** 2 fair
**Contribution:** 2 fair
**Rating:** 6
**Confidence:** 2

**Summary:**

The paper introduces image based visual servo to refine the coarse poses estimated using a coordinate regression network trained on neural rendering based 3D labels. The paper also uses correspondences in depth between neural rendered image and query image to iteratively refine the pose of the image. The paper also shows that NeRF-IBVS can be used as a navigation prior and subsequently use IBVS navigation.

**Strengths:**

1. The paper uses only a few images for visual localization compared to other methods out there
2. The paper does both visual localization and navigation based on visual servoing. The authors claim to be the first to do that.
3. The paper shows SOTA results on multiple datasets for both visual localization and navigation.
4. The paper provides and effective navigation prior which improves IBVS based navigation.

**Weaknesses:**

1. The paper trains the coordinate regression network using the approximate 3D labels obtained through neural rendering which is not ideal. A NERF trained with a small number of posed images has a lot of rendering noise, as discovered by the authors and shown in Fig 3.
2. It is not clear how the authors decide which pixels in the NERF-rendered images are used to train the coordinate regression net. They do mention that image boundaries are not used, but this is not a satisfactory description. Image boundaries from NERF may well be crisp, and parts of the image near the centre might have noise.
3. The coordinate regression network name is a bit confusing. At first glance, it appears as if it is a network that regresses pose of the camera from the 2D image. Actually, it is regressing 3D positions for each image pixel in the global coordinate frame.
4. The IBVS module is not clear. The authors suddenly introduce camera velocity and image velocity and a Jacobian without mentioning any prior information about this - which is presumably from the visual servoing literature. This material is provided in the supplementary, but should be included in the main paper.
5. The method is not real-time due to the time taken by neural-rendering, during the optimization step
6. The authors have not explored the joint training of NERF and the coordinate regression net, which might be an avenue to pursue.
7. The Quality of NERF rendering is crucial even for getting coarse poses that are subsequently used to train the coordinate regression net, and the PnP on the 3D poses and the 2D images. This will struggle with non-Lambertian and dynamic objects, etc.

**Questions:**

1. The paper could use better neural rendering techniques for latency and accuracy of depth and image rendering.
2. No details about the RANSAC algorithm used.
3. Can the superglue correspondence network used be made trainable?

**Limitations:**

1. The paper may not be able to handle instances where there are a lot of specular objects in the scene. This can lead to bad coarse pose and in turn IBVS may not converge.
2. The paper doesn't compare runtimes while comparing performance of other visual navigation methods. Some of the methods might have compromised on quality to reduce latency
3.The authors claim IBVS iterations are launched no more than 4 times. How was this arrived at ? Again this really depends on the quality of rendering. Any evidence to support this will help.

Typos:
-'Deired' -> Desired in caption for Figure 4.

---

> ### Author Rebuttal · Authors · 2023-08-09
>
> Thank you for the constructive comments and suggestions.
>
> **W1 and W2: The paper trains the coordinate regression network using the approximate 3D labels provided by NeRF which is not ideal. It is not clear how the authors decide which pixels in the NERF-rendered images are used to train the coordinate regression net.**
>
> Thanks for your insightful comment.
>
> 1) Since the pose optimization module will correct the coarse poses provided by the coordinate regression network with PnP. The coordinate regression networks only need to ensure that the estimated result is in the vicinity of the target, which does not need to be a very accurate result. Therefore the approximate 3D labels with rendering noise obtained through neural rendering are enough to train the coordinate regression network.
> 2) We empirically found that the rendering noise of NeRF mainly appears in the scene edge region (lack of multi-view constraints), which generally appear in the edge region of images, and the rendering noise in the center region of the images has little effect on the goal that the coordinate regression network only needs to estimate results near the target. Therefore, we empirically crop out information within 40 pixels from the boundary to minimize the effect of the rendering error.
> 3) To verify the above arguments, we perform quantitative experiments on the 12Scenes dataset. We consider that the coarse results provided by the coordinate regression network are in the vicinity of the target when the position and orientation error of the coarse pose reduces after pose optimization. Therefore, we count how often this happens, and the specific quantitative results are shown in the following table:
>
> | 12Scenes | kitchen1 | living1 | bed    | kitchen2 | living2 | luke   | gates362 | gates381 | lounge | manolis | 5a     | 5b     | Average    |
> |----------|----------|---------|--------|----------|---------|--------|----------|----------|--------|---------|--------|--------|--------|
> | **Frequency**     | 96.88%   | 99.59%  | 94.61% | 96.67%   | 95.42%  | 97.76% | 100.00%  | 95.16%   | 99.69% | 90.46%  | 90.14% | 91.60% | 95.66% |
>
> As shown in the table, the average frequency of coarse pose error reduction is greater than 95\% in all scenes. Therefore, the coordinate regression network consistently outputs results near the target despite only cropping the boundary information of the image.
>
> **W3: The coordinate regression network name is a bit confusing.**
>
> Thanks for your valuable suggestions.
> Following HACNet [20] and DSAC* [8] in the field of visual localization, which defines the coordinate regression network as regressing dense 3D coordinates directly from 2D images. We add the definition of coordinate regression networks in the methods section.
>
> **W4: The IBVS module is not clear.**
>
> Thanks for your valuable suggestions. We have rewritten the method section to make IBVS module clearer for readers and make this paper self-contained. Specifically, We add a subsection for the methods section called **“Preliminaries”** to introduce the core concepts of IBVS. Please refer to **Q2** in the **"General Response"** for specific modification details.
>
> **W5: The method is not real-time due to the time taken by neural rendering.**
>
> Thanks for your insightful comment.
> The efficiency is not the main problem addressed in the paper, we mainly take full advantage of the prior knowledge of the scene provided by NeRF to enhance visual localization and visual navigation.
> For example, in the 7Scenes dataset, the proposed method obtains better localization performance than the advanced real-time methods such as FeatLoc++Au [2] (our: 0.05m/1.55$^{\circ}$ vs FeatLoc++Au: 0.14m/5.89$^{\circ}$) and uses fewer posed images for supervision (our: 2090 vs FeatLoc++Au: 26000). Furthermore, our method enables navigation based on IBVS without using custom markers and the depth sensor compared to general IBVS-based navigation methods [9,17].
> In the future, we can try to use faster and more accurate variants of NeRF to improve the overall efficiency of the proposed method.
>
> **W6: The authors have not explored the joint training of NERF and the coordinate regression net.**
>
> Thanks for your valuable suggestions.
> The coordinate regression network needs the approximate labels provided by NeRF for training. Therefore, it is necessary to train NeRF first and then train the coordinate regression network. If the NeRF and coordinate regression network are trained jointly, it may be necessary to focus on the NeRF training in the early stage and focus on the coordinate regression network training in the later stage, which might be a worthy future direction.
>
> **W7: The Quality of NERF rendering is crucial. The proposed method will struggle with non-Lambertian and dynamic objects, etc.**
>
> Indeed, the quality of NeRF rendering is crucial for our method. A fundamental solution to this problem (caused by non-Lambertian and dynamic objects, etc.) requires further improvements in NeRF. In the future, the improved NeRF variants can be used to improve the performance of our method.
>
> **Q1: The paper could use better neural rendering techniques for latency and accuracy of depth and image rendering.**
>
> Theoretically, a better NeRF does lead to performance improvements, which is a good direction for future research.
>
> **Q2: No details about the RANSAC algorithm used.**
>
> Sorry for the unclear details. Actually, RANSAC is a common module in the field of visual localization. We have explained it in detail in **“1.4 Details of Pose Optimization and Correspondence Selection”** of the supplementary material.
>
> **Q3: Can the superglue correspondence network used be made trainable?**
>
> Yes, superglue can be trained, but generating the training labels requires a lot of manual annotations. Therefore, we do not train superglue in our dataset and only use its official weights.

---

> > ### Comment · Reviewer_iS7u · 2023-08-22
> >
> > Thank you for your rebuttal. You have answered some of my questions, but I maintain that the paper could be more clearly written. For example, I think you should motivate why exactly you need a separate coordinate regression network to get depth for the new rendered image? Don't you get the depth from NERF already? If it is because NERF depth is noisy, you should do an ablation study for using NERF depths to optimize the Visual Servoing.
> > I will maintain my previous rating.

---

> > > ### Author Response · Authors · 2023-08-22
> > > **Thank you for your comment!**
> > >
> > > We sincerely thank you for your efforts in reviewing our paper and your constructive suggestions again. We have added a lot of details to make our paper more clear to the readers. Please refer to Q2 in the "General Response" for major modification details.
> > >
> > > **The concern about the coordinate regression network:**
> > >
> > > Our method adopts a coarse-to-fine paradigm to estimate the pose of the query image. In the coarse stage, the coordinate regression network with pnp is used to provide the coarse pose for the query image. Specifically, the coordinate regression network inputs the query image (not the new rendered image) and then outputs its coarse 3D coordinates in the world coordinate frame. Finally, the coarse pose is obtained by pnp based on coarse 3D coordinates and 2D image coordinates.

---

> ### Comment · Area_Chair_Yfie · 2023-08-18
>
> Dear reviewer iS7u,
>
> could you please tell the authors (and us) whether your concerns have been answered?
>
> Best,
> AC

---

### Author Rebuttal · Authors · 2023-08-09

# General Response:
We thank all reviewers for their insightful and constructive suggestions, which help a lot in further improving our paper.

**Q1: The number of training data.**

We notice that several reviewers are concerned about the number of data required to train the proposed method compared to the baseline. We provide the specific number of training data as follows:

| 7Scenes  | Chess             | Fire               | Heads              | Office            | Pumpkin           | Kitchen           | Stairs             | All                |
|----------|-------------------|--------------------|--------------------|-------------------|-------------------|-------------------|--------------------|--------------------|
| **Baseline** | 4000(100\%)       | 2000(100\%)        | 1000(100\%)        | 6000(100\%)       | 4000(100\%)       | 7000(100\%)       | 2000(100\%)        | 26000(100\%)       |
| **Our**      | **260(6\%)** | **280(14\%)** |**240(24\%)** | **300(5\%)** | **320(8\%)** | **350(5\%)** | **340(17\%)** | **2090(8\%)** |

| 12Scenes | kitchen1   | living1     | bed        | kitchen2   | living2    | luke        | gates362    | gates381    | lounge     | manolis     | 5a          | 5b          | All          |
|----------|------------|-------------|------------|------------|------------|-------------|-------------|-------------|------------|-------------|-------------|-------------|--------------|
| **Baseline** | 744(100\%) | 1036(100\%) | 868(100\%) | 768(100\%) | 725(100\%) | 1370(100\%) | 3536(100\%) | 2950(100\%) | 925(100\%) | 1613(100\%) | 1001(100\%) | 1391(100\%) | 16927(100\%) |
| **Our**      | **185(25\%)**  | **170(16\%)**   | **215(25\%)**  | **190(25\%)**  | **180(25\%)**  | **340(25\%)**   | **350(10\%)**   | **295(10\%)**   | **230(25\%)**  | **270(17\%)**   | **250(25\%)**   | **280(20\%)**   | **2955(17\%)**   |

Actually, we have included these tables in the section **"1.1 Amount of Training Data for Each Scene"** of the supplementary material. And we will include these tables in the paper to make it clear to readers.

**Q2: The writing issue.**

We notice that several reviewers suggest that the paper writing could be further polished. Therefore, we have worked on both readability and language and have also involved native English speakers for language improvement. We also added a lot of technical details to make our paper more understandable to the readers. We list the major changes as follows:

We have rewritten the method section to make IBVS module clearer for readers and make this paper self-contained. Specifically, we add a subsection for the methods section called **“Preliminaries”** to introduce the core concepts of IBVS: “The aim of Image-Based Visual Servoing (IBVS) [9] is to control the camera to move towards the desired pose based on vision information while minimizing the error of correspondences between the current image and the target image. To achieve this goal, IBVS first calculates the desired image coordinate velocity of correspondences based on the coordinate error of correspondences. Then, the Jacobi matrix is constructed based on the correspondences which establishes the relationship between the image coordinate velocity and the camera velocity. Finally, the desired camera velocity to control the camera motion towards the target is solved based on the Jacobi matrix and the desired image coordinate velocity. The specific details of the IBVS are presented in the supplemental materials”.

We add details and fix typos in the paper. We add "M is the number of points along the ray" for Eq1. We add the explicit mention in L201: "After that, the correspondence selection algorithm can obtain accurate and non-collinear correspondences and the IBVS module can obtain navigation trajectories. Therefore we can obtain accurate and non-collinear correspondences with navigation trajectories".
We add “$e(x_i), e(y_i)$ is the image coordinate error of correspondences” and correct “Deired” to “Desired ” in Fig4.
We update "only a few images" to "fewer images" in L7.
We update "To achieve this, we first use NeRF to generate pseudo-3D labels which are used to train the scene coordinate regression network. Then a coarse pose is estimated from the regression network. Finally, we use the image-based visual servo (IBVS) to utilize 3D scenes provided by NeRF for pose optimization" in L8.
We update "Then, we establish the correspondences between the rendered image and the query image and query the approximate depth of the correspondences based on the rendering depth map. Finally, the correspondences with depth are used to launch IBVS to guide pose optimization" in L53.
We update "where $x, y$ and $Z_c$ denote image coordinates and depth of correspondences in the currently rendered image" in L141.
We add "Navigation prior is accurate and non-collinear correspondences and navigation trajectories corresponding to the correspondences" in L61.
We rewrite the "geometric constraints" to "multi-view constraints" which are the basic concepts in multi-view stereo in L125.
We rewrite the "edge areas" to "edge areas in the image" in L127.
We update "Due to rendering errors of NeRF, the assumption that the 3D coordinates of correspondences are accurate is violated so accumulation errors will occur" in L168.
We correct “enables” to “enable” in  Fig 1 and L 62. We correct “We” to “we” in L 103. We add  "$t_n$ is the near bound of the camera ray" in L 116.

---

### Decision · Program_Chairs · 2023-09-21

**Decision:**

Accept (poster)

**Comment:**

This paper proposes a method for visual localization which generates images and 3D labels with a NERF, followed by visual servoeing.

Overal, the paper had received 4 positive out of 5 ratings and was disscussed by reviewers and authors and then reviewers and AC. The reviewers appreciated novelty, strong results, the usage as a navigation prior. Issues raised were on clarity (the main issue) and some minor discussions happened on comparisons with differentiable baselines like iNERF. The rebuttal could answer many of the questions. Although some issues on clarity remained, the AC judges the paper to be ready for publication.